# Mediating Role of Intra-Team Conflict between Paternalistic Leadership and Decision-Making Quality among China University’s CMT during COVID-19

**DOI:** 10.3390/ijerph191811697

**Published:** 2022-09-16

**Authors:** Kenny S. L. Cheah, Zuraidah Abdullah, Min Xiao

**Affiliations:** 1Department of Education Management, Planning and Policy, Faculty of Education, University Malaya, Kuala Lumpur 50603, Malaysia; 2School of Public Finance & Public Administration, Jiangxi University of Finance & Economics, Nanchang 330013, China

**Keywords:** crisis management teams (CMTS), paternalistic leadership, intra-team conflict, decision-making quality, COVID-19 crisis management

## Abstract

Universities across China have set up crisis management teams (CMTS) to deal with the crisis brought on by the COVID-19 pandemic. This study focuses on how the paternalistic leadership practices of a Chinese university CMT influence crisis strategic decisions by managing conflict. These relationships were verified using hierarchical regression analysis on 312 samples from the surveyed university during the pandemic and found the following: benevolent leadership and moral leadership have positive effects on decision quality. However, unlike most studies on paternalistic leadership, in crisis situations, the negative effects of authoritarian leadership disappear under the mediating effect of affective conflict. This means that affective conflict within CMT fully mediates the relationship between authoritarian leadership and decision quality, and partially mediates the relationship between moral leadership and decision quality, while cognitive conflict partially mediates the relationship between benevolent leadership and crisis decision quality. It indicates that a CMT must stimulate and maintain a certain level of cognitive conflict while suppressing affective conflict to achieve high-quality crisis decision-making. This state can be achieved by practicing lower levels of authoritarian leadership and maintaining high levels of moral and benevolent leadership practices.

## 1. Introduction

The COVID-19 pandemic is the world’s most serious biological threat of the 21st century [1]. The pandemic has changed the whole world and also has brought great crises to universities [2]. Since March 2020, universities in almost all countries around the world have begun to cancel face-to-face lectures or even close [3], but in some countries and regions, universities are actively responding and changing to try to find a way out of the pandemic threat.

Although experiencing wave after wave of pandemic shocks, Chinese universities are gradually returning to normal teaching [2]. In addition to adhering to the government’s instructions, Chinese university administrators also face higher management challenges, especially how to deal with the crisis. In response to the pandemic, almost all Chinese universities have set up crisis management teams (CMTS) to integrate different resources and manpower on campus. In this context, the decision-making mode of Chinese university leaders is also changing, from the traditional small decision-making team to the sharing and participation of large strategic decision-making teams.

Leadership plays a crucial role in crisis responses [4]. Leadership plays a decisive role in stimulating the creativity of decision-making teams, as well as in sharing information, unifying cognition, and jointly making consistent decisions [5,6,7]. Correspondingly, intra-team conflict is also an important process factor in the decision-making process [8]. The larger the group, the more likely it is to be divided, leading to more conflicts.

In the CMT process of Chinese universities, a leadership with distinct Chinese Confucian traditional culture is reflected: paternalistic leadership (PL) [9,10]. This is a kind of integrated leadership that includes three dimensions: authoritarianism, morality, and benevolence [11]. At the same time, conflict within the decision-making team also shows a very complex role. In addition to the negative impact commonly understood by people, some researchers have found that conflict can have a positive impact on decision-making under special circumstances [12,13,14,15]. Based on the research of Jehn [16,17], Amason [18] divided this kind of conflict into two categories, cognitive conflict and affective conflict, believing that cognitive conflict is a kind of functional conflict, which may have a positive impact on the decision-making effect, while affective conflict is a kind of destructive conflict, which will harm the decision-making effect.

Based on these findings, it is necessary to research how educational institutions are coping with the COVID-19 crisis in Chinese universities, especially with regards to the role of paternalistic leadership practices in the decision-making processes of university CMTs. On the other hand, due to the importance of conflict management in crisis responses, it is necessary to study its role in the team process to further understand the role path of crisis decision-making.

## 2. Literature Review

### 2.1. Crisis Decision-Making Quality in Chinese Universities

The main decision-making team of a university consists of the administrative committee and the academic committee, which are respectively responsible for the strategic planning, structural decisions, and teaching–research decisions of the university [19]. On this basis, Chinese universities have added Party committees, which are responsible for representing the political power [20]. Therefore, under the general collegial system, another hierarchical system exists from top to bottom in Chinese universities. Research on the performance of this complex work system under the pandemic crisis is necessary to further understand the crisis management mechanism in Chinese universities.

To better cope with the crisis, but also to better integrate the two systems, the university CMT would act as a suitable temporary response institution. A CMT should include multiple departments to respond to internal and external crises more effectively and quickly [21]. The crisis decision-making ability of CMT determines whether the campus can safely survive a crisis. Different from general decision-making, crisis decision-making is a semi-structured or unstructured process subject to limited time, information, and additional pressure [22], which is undoubtedly a huge challenge for every leader in CMTs.

However, it is difficult to measure the effect of group decision-making with objective indicators at present, and most studies can only judge based on the perception and subjective evaluation of involved decision makers, such as their quality, understanding, commitment, and affective acceptance [18]. Decision-making quality is directly related to organizational performance, understanding and commitment determine the effect of decision implementation, and affective acceptance reflects the emotional relationship between decision-making team members [18,23].

It is precisely because the CMT process in universities often faces a complex and uncertain environment that the quality of the final decision is often affected by many factors. Based on a review of past literature, this study believes that CMT leadership behavior is an important factor affecting the quality of crisis decision-making, which has been confirmed by many studies [24,25,26,27]. Intra-team conflict is another factor that has a further impact on crisis quality. Due to the multi-dimensional nature of conflict, its different dimensions have inconsistent impacts on crisis quality [18,23].

Universities, as institutions that play an important educational role in society, and as places with high personnel density and mobility, will be greatly affected by the pandemic. However, there is a lack of research on the response of university management to the COVID-19 crisis. Based on the past literature, this study constructed a CMT process model for universities and conducted empirical research to try to explain the relationship between the influencing factors surrounding the quality of crisis decision-making.

### 2.2. Paternalistic Leadership

House et al. [28] carried out a large-scale cross-regional study “GLOBE” (Global Leadership & Organizational Behavior Effectiveness) and found that leadership behavior and effectiveness are not only influenced by regional cultural differences [29,30] but can also be influenced by individual cultural tendencies. After a series of studies, scholars have found that there exists a leadership style similar to transformational leadership, but with distinct dialectical and unified characteristics, paternalistic leadership (PL), among the leaders of Chinese organizations. This is a kind of complex leadership behavior describing a strong authority style similar to paternity, caring and caring for subordinates, and showing a high moral self-integrity [11].

The reason why paternalistic leaders show such a complex leadership style is closely related to the Confucian thoughts of paternalism and man-rule under the traditional Chinese culture. Redding and Hsiao found that in the patriarchal atmosphere, organizations would show strict hierarchy and vertical obligation relationships [31]. However, under the influence of man-rule, the establishment of interpersonal relations in the organization reflects another kind of control: loyalty and obedience are exchanged by favoritism toward subordinates [11].

For thousands of years of history, Chinese people have relied on family members to create and manage organizations and to survive in a society in the form of “family organization” [32]. Partly due to the long-term immersion of Confucian culture and the influence of the imperial system, Chinese people not only expect the parents of family organizations to be strong leaders but also expect the leaders of all organizations to play the role of “parents” [33]. Although Western researchers have different descriptions of this kind of paternalistic leadership, the definition proposed by Farh and Cheng is generally accepted by scholars: a leadership style that combines strong discipline and authority, paternal kindness, and moral character [11]. Paternalistic leaders constantly guide and educate their subordinates, maintain a certain social distance from them to build prestige, expect their subordinates to follow them by showing a benevolent side, and try to maintain a noble image.

Farh and his colleagues put forward a model of paternalistic leadership, which includes three dimensions: authoritarian leadership (AL), benevolent leadership (BL), and moral leadership (ML) [11], believing that the values and ideologies of traditional Chinese culture are hidden under these three dimensions. However, scholars’ understanding of the internal constructs and effectiveness of paternalistic leadership has not been unified so far [34,35].

Lin et al. found that from the perspective of gender role theory, the triplet behavioral dimensions of paternalistic leadership exactly represent male-oriented (authoritarian leadership), female-oriented (benevolent leadership), and gender-neutral (moral leadership) characteristics [34]. Therefore, they are completely different from each other, which has led a large number of scholars to generally believe that authoritarian leadership harms organizational outcomes, while benevolent leadership and moral leadership have a positive impact on organizational outcomes [36,37,38]. Some scholars found that under certain circumstances, the effectiveness of the three dimensions of paternalistic leadership could reach a consensus [23].

In addition, Alsamaray studied the relationship between crisis management and leadership style and found that under crises, leadership with an authoritarian style often has strong effectiveness, while leading with a democratic style has a certain correlation with crisis management [39]. While Wang studied the paternalistic leadership of rescue teams in crises brought about by earthquakes, he also found that authoritarian leadership had a positive impact on team results in such specific situations, which was inconsistent with the findings of most scholars [40].

Since this study was also conducted in the crisis context brought by the COVID-19 pandemic, it is necessary to further verify the practice of paternalistic leadership theory in the crisis context, especially in the educational context, to fill the research gap in this aspect.

### 2.3. Intra-Team Conflict

Conflict is an integral part of human activity and has existed in various forms throughout human history. Most theoretical scholars acknowledge the importance of conflict, but so far it has not been well understood and explained, especially regarding the conflict in the process of group decision-making, and there are still many contents to be further studied [41].

When compared with the ordinary management situation, conflict management in a crisis is particularly important: the increased uncertainty internal and external factors, the limited decision-making time, and the accumulation of pressure inside and outside the organization all have interaction effects in the process of CMT, which is manifested in various conflict forms and further affects the quality of crisis decision-making. According to McDonald, nothing is more dangerous for an organization than conflict between senior leaders, which means inefficiency and pointless internal friction [42]. Under his influence, a large number of scholars believe that conflict is harmful to the organization, decision-making, and effectiveness [17,43,44,45]. However, it is now being gradually recognized that conflict, when properly used, can be beneficial to the innovation and competitiveness of teams and organizations [18,46,47,48].

In addition, the classification and definition of intra-team conflicts have not been unified so far [49]. According to the antecedent conditions of conflict, Jehn and other scholars divided intra-team conflict into two categories, one related to the task (cognition) and the other related to the relationship (emotional) [16,17]. According to Jehn, “people tend to dislike people who disagree with them or who don’t share their values and beliefs [17] (p. 258)”. Therefore, she defines affective conflict as “conflict caused by the realization of disharmony between people by all parties involved [17] (p. 258)”. At the same time, she also made it clear that conflicts can also occur due to differences in opinions or viewpoints and defined cognitive conflict as “conflict caused by the inconsistent understanding of the century-long task being carried out by the parties involved [17] (p. 259)”. Some scholars have supported these definitions [18,50].

Since the research on the team process is similar to exploring the “black box” [51], scholars are constantly exploring what factors affect team results. Many scholars have confirmed that the two-dimensional nature of intra-team conflict has different effects on team outcomes. In general, scholars have a relatively consistent understanding of the mechanism of affective conflict and have found that it affects team productivity or creativity [52,53,54], employee satisfaction [55], and decision-making quality and effectiveness [18,56] with negative consequences.

However, the role of cognitive conflict remains contradictory. Many scholars have proved that cognitive conflict harms both decision-making quality and team performance, but it can have a positive impact under special circumstances [12,13,14,15]. For example, cognitive conflict is conducive to promoting divergent thinking and avoiding team thinking, which is important for creative problem solving and team performance [57]. In the research on the mechanism of intra-team conflict, Amason found that cognitive conflict can improve the decision-making effect, while affective conflict is harmful to the decision-making effect [18]. At the same time, because the improvement of the cognitive conflict level will lead to the improvement of affective conflict, the different strengths of this interaction may lead to a difference in the effect of decision-making quality [18,56].

Based on these findings, to improve the effectiveness of crisis response, CMTs are inevitably faced with conflict management in crises. This conflict management is not simply used to reduce conflicts but to eliminate, as much as possible, the affective conflicts that are harmful to the decision-making process, while maintaining the cognitive conflicts that are beneficial to the decision-making process.

### 2.4. Paternalistic Leadership and Crisis Decision-Making Quality

In the process of reviewing the literature on paternalistic leadership, it was found that there are few empirical studies on the practice of paternalistic leadership in educational institutions. Studies in this theme compare paternalistic leadership with different leadership models (e.g., transformational leadership) in the school context [58]. There is also some research on the paternalistic leadership practices of sports coaches [59]. However, there are few studies on other aspects of educational institutions, especially on university presidents [34,35].

In the educational environment, leaders are often task-oriented, aiming at improving students’ performance or teaching quality [60], while paternalistic leadership focuses on cultural environment and relationship management [11,23,61]. As a result, schools that practice paternalistic leadership are more likely to discuss emotional responses or interpersonal relationships brought about by such leadership practices [23,62]. However, at the same time, it has been confirmed by many scholars that it is a common leadership style in Chinese schools [10,63,64]. Therefore, the influence of this leadership style on group decision-making under crises is worth discussing.

The authoritarian leadership dimension in paternalistic leadership usually requires the absolute obedience of subordinates, which leads to the improvement of internal pressure and even negative emotions in the decision-making team [65], and changes from collective decision-making to individual decision-making thus lead to the decline of a team’s decision-making quality [66]. The benevolent leadership dimension promotes the improvement of decision-making quality [23,67] by creating a supportive atmosphere [68] and even favoring subordinates [69]. The dimension of moral leadership describes obtaining the recognition of other team members by setting an example and showing a noble moral model [11], which promotes team cooperation and improves the quality of decision-making [23,61,70].

Based on the above findings, this study puts forward the following research hypotheses:

**H1a.** *Authoritarian leadership practices of CMT members are negatively related to crisis decision-making quality*.

**H1b.** *Benevolent leadership practices of CMT members are positively related to crisis decision-making quality*.

**H1c.** *Moral leadership practices of CMT members are positively related to crisis decision-making quality*.

### 2.5. The Mediating Role of Intra-Team Conflict

Scholars describe the team process according to the input–output system model and find that leadership is a crucial variable in the input factors [71,72]. It can be said that no group decision-making process is not influenced by the leadership [41]. Intra-team conflict, as discussed above, is another core variable in the team “black box” that may have a direct impact on team outcomes [56].

Benevolent leadership in paternalistic leadership emphasizes harmonious working relations, understanding, and tolerance of subordinates, which can create an open team division and enhance cohesion among members [23,73], contributing to cognitive conflict in teams [74]. At the same time, benevolent leadership is also beneficial to improving interpersonal relationships within the team, promoting loyalty and trust [75,76], and these variables are significantly negatively correlated with affective conflict [77].

Therefore, this study proposes this hypothesis:

**H2a.** 
*CMT members’ benevolent leadership practice is positively correlated with cognitive conflict within the team, and negatively correlated with affective conflict.*


Authoritarian leaders are widely believed to play a negative role in team processes. Leaders with high authoritarianism often show a strong desire to control the decision-making trend, closely control the communication among members, intercept the key information related to decision-making [23], and insist on making decisions by themselves. This tends to stimulate more negative emotions within the team, which can trigger levels of affective conflict. At the same time, because this behavior is not conducive to communication and interaction among members, it will further reduce the level of cognitive conflict within the team. Based on this, this study proposes this hypothesis:

**H2b.** *CMT members’ authoritarian leadership practices are negatively correlated with intra-team cognitive conflict, and positively correlated with affective conflict*.

The most important characteristics of moral leadership are fairness and justice and leading by example [11]. Therefore, whether CMT members can give up their selfishness within the team process depends on the level of their moral leadership. High-level moral leadership practice has been proven to enhance the mutual benefit of the whole team [23,78], which in turn greatly reduces affective conflict within the team [26,78]. Becker believes that honesty and trustworthiness are the most important characteristics of moral leaders, which will help them win the trust and respect of followers [79]. Trust has been proven to be a positive factor in improving cognitive conflict [18]. Therefore, this hypothesis is proposed:

**H2c.** *Moral leadership practice of CMT members is positively correlated with cognitive conflict within the team, and negatively correlated with affective conflict*.

Scholars believe that conflict is mainly caused by the difference in values caused by the conflict between resource scarcity and individual needs [17,80]. Wall and Canister believed that conflicts were mainly caused by individual factors and factors between individuals [81]. At present, there is still no conclusion on exactly which factors cause conflict in teams, but scholars have found that factors related to leadership appear in these antecedent variables, such as trust [77,82], values [47], cohesion [48], communication, goals [81], etc.

These findings suggest that the leadership behavior and style of decision-making of the team members may influence the level of intra-team conflict. Ayoko and Chua found that leadership is the antecedent variable of intra-team conflict [83], and Balkundi et al. also obtained the same conclusion [84]. As described above, a large number of scholars have confirmed the relationship between different dimensions of intra-team conflict and decision quality [18,56,85]. In conclusion, intra-team conflict plays an important mediating role between team leadership and decision quality.

Therefore, the following hypotheses are proposed in this study:

**H3a.** *Cognitive conflict mediates the relationship between paternalistic leadership practices and decision-making quality in university CMT*.

**H3b.** *Affective conflict mediates the relationship between paternalistic leadership practices and decision-making quality in CMT*.

According to the above research hypotheses, this study proposes the theoretical and conceptual framework as shown in Figure 1.

## 3. Methods

### 3.1. Research Environment

The impact of the pandemic has made it extremely difficult to collect data for surveys. Because of movement restrictions, researchers are unable to move freely between China’s multiple universities to collect data. In addition, due to the consistency of the university system in China, universities in the same region often face the same external environment and challenges, and their internal management is also highly similar [20]. Therefore, this study decided to select one of several universities in a certain region of China for empirical research. The surveyed university meets the following conditions: 1. Distance. The surveyed university is located within 500 km of Wuhan, which harbored the first incidence of the COVID-19 pandemic and is a high-risk area. 2. Response effect. The surveyed university has had no infection cases since 2019. 3. Representativeness. The surveyed university is a public university with the same management structure and scale as most universities in China. 4. Accessibility. After the early efforts of the researchers, this study was approved and supported by the leaders of the surveyed university to engage in data collection on a campus scale.

### 3.2. Sample

The study was conducted at the university during a new round of the COVID-19 pandemic in southeastern China in March 2022. To avoid the influence of common variance, each CMT member’s paternalistic leadership practices were evaluated by their direct subordinates, meanwhile, the intra-team conflict and decision-making quality were evaluated by CMT members. The researchers took several measures to make sure the data were matched. All participants in this survey were anonymous and voluntary.

The CMT of the surveyed university consists of 33 members, 32 of whom, as well as their subordinates, participated in the survey, and 312 pairs of samples were finally obtained. Each CMT member had no less than 3 subordinates who participated in the survey. All 32 CMT members were male, 78.1% of whom had a master’s degree or above and had different professional backgrounds. In total, 18 of them are from administrative positions, 7 are deans of schools, 4 are heads of research institutions, and 1 each are involved in logistics, security, and medical departments. Among these university elites, 23 have more than 10 years of leadership experience, among which 9 have more than 20 years of leadership experience.

Among the subordinates of CMT members who participated in the survey, 68.9% were female, all of whom had a bachelor’s degree or above, and 5.8% had a master’s degree or above.

### 3.3. Measure

#### 3.3.1. Decision-Making Quality

The quality of CMT crisis decision-making was measured using the Decision effect scale used by Amason [18]. The scale has 4 dimensions, which are “Quality”, “Understanding”, “Commitment”, and “Affective Acceptance”. In this study, the “Quality” dimension was selected to measure the decision-making quality, including “the overall quality of the decision”, “the quality of the decision relative to its original intent”, and “The quality of the decision given its effect on organizational performance”. This subscale is scored with a 4-point Likert scale ranging from 1, “poor”, to 4, “excellent”. The reliability coefficient was 93.

#### 3.3.2. Paternalistic Leadership Practices

The paternalistic leadership practices of CMT members of the surveyed university were measured using a simplified scale adapted by Farh et al. [86]. The scale includes three dimensions: benevolent leadership (6 items), moral leadership (4 items), and authoritarian leadership (9 items), with a total of 19 items. This is a 5-point Likert-type scale, with one endpoint 1, “Strongly Disagree”, to the other endpoint 5, “Strongly Agree”. The reliability range is 93–95, and the overall reliability coefficient is 80.

#### 3.3.3. Intra-Team Conflict

The cognitive and affective conflicts of CMT members were measured by Chen et al. [23] based on the simplified Chinese version of the intra-team conflict scale developed by Amason [18] and Jehn [16]. Three items were used to measure cognitive and affective conflicts respectively, with a total of 6 items. The scale is measured by a 5-point Likert-type scale, with one endpoint 1, “Strongly Disagree”, to the other endpoint 5, “Strongly Agree”, the reliability coefficients are 87 and 93 respectively, and the overall reliability coefficient is 83.

#### 3.3.4. Control Variables

In this study, demographic variables such as CMT members’ level of education, professional background, length of service, and the type of position that may have an impact on team results were considered as control variables. In addition, because the paternalistic leadership practices of CMT members come from the evaluation of subordinates, demographic variables such as subordinates’ gender and level of education were also included in the model as control variables.

### 3.4. Data Analysis

All research instruments were pilot tested before the actual study, and all non-Chinese versions of research instruments were back-translated by experts who were proficient in both Chinese and English to ensure their use in the Chinese context. Table 1 lists the results of the validity and reliability analyses of the research instruments. This study used SPSS 26 and SPSSAU to conduct descriptive statistics and hierarchical linear regression analysis, respectively, to verify the research hypothesis proposed in this study.

## 4. Results

### 4.1. Descriptive Statistics & Correlation Analysis

The means, standard deviations, and correlations of all the study variables are shown in Table 2.

CMT members of the surveyed university rated the quality of crisis decision-making as generally good during the COVID-19 pandemic (M = 2.44, SD = 0.82, Max = 4). Meanwhile, during the crisis management decision-making members practiced general authoritarian leadership behavior (M = 2.70, SD = 1.16), high moral leadership behavior (M = 3.81, SD = 0.79), and benevolent leadership behavior (M = 3.82, SD = 1.02). In the process of decision-making, a high level of cognitive conflict (M = 3.38, SD = 0.58) and a general level of affective conflict (M = 2.46, SD = 1.03) were generated.

According to the correlation analysis, the benevolent and moral dimensions of CMT’s paternalistic leadership practice were significantly positively correlated with the quality of decision-making during the COVID-19 crisis (*r*1 = 0.60, *p* < 0.01; *r*2 = 0.58, *p* < 0.01), while the dimension of authoritarian leadership was significantly negatively correlated with decision quality (*r* = −0.38, *p* < 0.01). The cognitive conflict was positively correlated with the decision quality during CMT intra-team conflict (*r* = 0.41, *p* < 0.01), while the affective conflict was negatively correlated with decision quality (*r* = −0.26, *p* < 0.01). There was also a significant positive correlation between cognitive conflict and affective conflict within the team (*r* = 0.14, *p* < 0.05).

The relationship between CMT members’ different paternalistic leadership behaviors and conflict is also different. Among them, authoritarian leadership practices were significantly negatively correlated with cognitive conflict (*r* = −0.33, *p* < 0.01) and was positively correlated with affective conflict (*r* = 0.23, *p* < 0.01). The practices of benevolent leadership and moral leadership with cognitive conflict were significantly positively correlated (*r*1 = 0.47, *p* < 0.01; *r*2 = 0.53, *p* < 0.01) and significantly negatively correlated with affective conflict (*r*1 = −0.35, *p* < 0.01; *r*2 = −0.28, *p* < 0.01), respectively.

### 4.2. Hierarchical Regression Analysis

To further verify the research hypotheses, a hierarchical regression analysis was conducted in this study, and the results are shown in Table 3.

After controlling for demographic variables, the three dimensions of paternalistic leadership have different effects on decision-making quality. Among them, moral leadership and benevolent leadership had a significant positive potential impact on the quality of crisis decision-making (*β*1 = 0.18, *p* < 0.01; *β*2 = 0. 08, *p* < 0.05), while the potential negative impact of authoritarian leadership on the quality of crisis decision-making disappeared after the introduction of intra-team conflict (*β* = −0.05, *p* < 0.05). H1b and H1c are supported by the data, while H1a is not.

Among the three internal dimensions of paternalistic leadership, only benevolent leadership had a significant positive impact on cognitive conflict (*β* = 0.26, *p* < 0.01); authoritarian leadership had a significant positive effect on affective conflict (*β* = 0.19, *p* < 0.01) and moral leadership had a significant inhibitory effect on affective conflict (*β* = −0.20, *p* < 0.01). H2a, H2b, and H3b are partially supported by the data respectively.

Model 4 shows that cognitive conflict has a significant and strong positive impact on the quality of crisis decision-making (*β* = 0.41, *p* < 0.01), and affective conflict has a significant negative impact on the quality of crisis decision-making (*β* = −0.23, *p* < 0.01). Combined with the above findings, the mediating effect was further analyzed by calculating the mediating effect size, and the results in Table 4 were obtained.

The results show that affective conflict fully mediates the relationship between the authoritarian leadership dimension and crisis decision quality of paternalistic leadership (100% effect), and partially mediates the relationship between the moral leadership dimension and crisis decision quality (17.75% effect). Cognitive conflict partially mediates the relationship between the benevolent leadership dimension and crisis decision-making quality, accounting for 54.32%. Both H3a and H3b are partially supported by the data.

## 5. Discussion

The findings of this study are similar to the conclusions obtained by most scholars. Different dimensions of paternalistic leadership practice have inconsistent effects on team outcomes (crisis decision-making quality). Generally speaking, benevolent leadership and moral leadership have positive potential effects on team outcomes. On the other hand, authoritarian leaders are generally believed to be negative about team results [62,64,75]. However, this study also found that this negative effect of authoritarian leadership disappeared as a mediator in the paternalistic leadership practices of the surveyed university CMT in a crisis (the COVID-19 pandemic). Similar to the findings of Chen et al. [23], this study also proves that the contradictions of different dimensions within paternalistic leadership can be reached under special circumstances, and intra-team conflict plays an important mediating role in the CMT process.

### 5.1. Paternalistic Leadership Practices and Decision-Making Quality of University CMT in Crisis

According to the results of descriptive statistics, the CMT of the surveyed university has different degrees of leadership practice in different dimensions of paternalistic leadership in the process of coping with the COVID-19 pandemic. This indicates that the organizational structure of Chinese universities represented by the surveyed universities has undergone a great modernization reform, especially in the decision-making team of the universities, and the influence brought by this reform is particularly important [87].

#### 5.1.1. Moral Leadership Practice and Decision-Making Quality

The results of this study prove that moral leadership plays a central role in the CMT process among the three dimensions of paternalistic leadership. In both the direct effect on decision quality and the indirect effect with the introduction of mediating variables, the absolute value of the standardized regression coefficient of moral leadership is the highest relative to the other two dimensions. In its long history, China has been a society that advocates “rule by man”. Under the interactive influence of high power distance and particularism, there is no effective guarantee for the formulation and implementation of laws and institutions. Therefore, people tend to expect their leaders to have a high sense of morality to avoid their abuse of authority or power relations [11,32].

According to Farh et al., without the moral constraints of the leader’s ego, authoritarian leadership and benevolent leadership will be reduced to the politics of the leader, which is used to manipulate or even sacrifice subordinates to extract their interests [69] (p. 191). In the CMT process of the investigated universities, moral sense urges CMT members to curb harmful behaviors based on their personal desires and prevent the phenomenon of “organizational goal substitution” that puts personal goals above organizational goals, thus ensuring that the CMT can jointly make high-quality decisions that truly represent the collective interests. At the same time, such a high moral atmosphere can improve CMT members’ perception of decision-making justice and make it easier to achieve cognitive consistency, thus promoting the improvement of decision-making quality [18].

#### 5.1.2. Benevolent Leadership Practices and Decision-Making Quality

In a crisis, the decision-making team of the university has changed from a small single-level decision-making team of the president, who usually implements the collegial system, to a medium-large crisis management team with a multi-level and multi-functional background. In this case, the power distance between CMT members has become smaller, and the principal (or the secretary of the party committee), though still playing the key decision maker in the group, has to practice more benevolent leadership behavior to build a relatively equal and easy decision environment, in order to better cope with the uncertainty of the external environment crisis. Such an environment facilitates intra-team communication and promotes diverse perceptions, thus facilitating the decision-making team to make higher-quality decisions together.

The findings of this study corroborate the findings of Pellegrini et al.: that benevolent leadership has a positive influence in cultures with a low power distance [88]. With the advancement of the modernization reform of higher education in China, the management model of Chinese university leaders also conforms to the development model proposed by Schein, that is, the transition from authoritarianism to familism, and finally to the modern leadership model combining participative and instructional leadership [89]. Benevolent leadership, as an important element of paternalistic leadership, combines the characteristics of participative leadership, instructional leadership, and transformational leadership, and plays a positive role in the increasingly equal political environment emphasized by Chinese universities.

#### 5.1.3. Authoritarian Leadership Practice and Decision-Making Quality

Meanwhile, under the threat of the COVID-19 pandemic, the negative impact of authoritarian leadership on the decision-making quality of CMT members of the surveyed university disappeared under the influence of mediating effects. At the same time, this study found that CMT members scored low on the level of authoritarian leadership practice. This shows that the value orientation of authoritarianism in paternalistic leadership is weakened in modern Chinese universities. As young people become more educated, teachers and students become more resistant to bureaucratic hierarchies. Moreover, the CMT members’ high education also makes the team’s acceptance of authoritarian orientation values not high, which leads to a low degree of authoritarian leadership practices in the process of CMT.

Additionally, as the entire school experiences the COVID-19 pandemic, CMT members face a common threat from the outside. This common threat forces CMT calendar personnel to be less willing to rely on senior leaders and to no longer passively accept and follow unreasonable orders or instructions, and thus more willing to take responsibility. On the other hand, due to the particularity of the Chinese system, the responsibility pressure from the regional authorities at the higher level of the university has also further compressed the power distance within the university’s CMT so that each CMT member has shared these external pressures, and more emphasis has been placed on individual participation and contribution rather than individual authority.

### 5.2. The Mediating Role of Intra-Team Conflict in University CMT Process

This study focused on examining the mediating role of intra-team conflict in the CMT process. It was found that cognitive conflict among CMT members of the surveyed university only partially mediates the relationship between benevolent leadership practices and decision quality in specific crises. Affective conflict also plays a part in mediating the relationship between moral leadership practice and decision quality. At the same time, this study also found that authoritarian leadership indirectly negatively impacts decision-making quality by stimulating or worsening affective conflict.

A benevolent leader enhances decision-making by providing a relaxed and secure environment for discussion and encouraging other members to fully express their views, providing diverse information for the CMT process. Falk found that one of the necessary conditions for cognitive conflict is this secure environment [74]. Therefore, the benevolent leadership practiced by CMT members in a common crisis can create an environment conducive to cognitive conflict [74], thereby enhancing the cognitive diversity of members, promoting information exchange within the team, enhancing cooperation among members, and thus realizing the improvement of collective decision-making quality [18,56].

This study found that moral leadership plays a central role in the paternalistic leadership practices of CMT, especially in the university environment under crises, where the moral character of CMT members appears to be more important. The findings of this study indicate that moral leadership has a significant inhibitory effect on affective conflict. This shows the importance of fairness and justice of leaders in creating a harmonious atmosphere, which can suppress and reduce interpersonal conflicts, that is, control affective conflicts within the organization and reduce their potential adverse impacts on team decision-making.

The main source of affective conflict can be attributed to the authoritarian leadership behavior in the team. Among the leadership practices of CMT in the surveyed university, only authoritarian leadership significantly affected affective conflict. affective conflict has been proved by a large number of studies to be detrimental to the improvement of decision-making quality [18,23,40,56] and this study also obtained the same result; after controlling for other variables, affective conflict still had a significant inhibitory effect on decision-making quality. This destructive, dysfunctional conflict pattern, in turn, leads to poor quality decisions.

## 6. Conclusions

The conclusions of this study explain the CMT process in universities in the context of Chinese culture, while also extending the literature on paternalistic leadership in the educational field. This study also provides a new explanation for the mechanism of leadership theory in specific crisis situations and explains the contradictions in the internal triad structure of paternalistic leadership, which to some extent fills the theoretical gap.

According to the theoretical and conceptual framework proposed in this study, it can be found that even if the CMT process is conceptualized as a model of paternalistic leadership–intra-team conflict-decision quality, it is still a very complex and delicate process [90]. Although there are internal contradictions in the internal dimension, paternalistic leadership is still a comprehensive and three-dimensional leadership theory, which can well explain the behavior of leaders in the context of Chinese culture.

Over the past 70 years since the founding of the People’s Republic of China, China has experienced an economic system reform from a planned economy to a fully open economy. Under this social background, people’s expectations of organizations have also undergone tremendous changes in a short period. At first, people were just satisfied with how to achieve tasks, interpersonal harmony, voluntary allocation fairness, and so on [32]; as the process of globalization and modernization has been occurring, now China’s organization managers need to focus on strategic planning and decision-making, not just need, through a constant adjustment in order to adapt to the rapidly changing external environment.

The pandemic has only accelerated that change. For Chinese universities, this drastic change in the external environment has changed the decision-making form and leadership behavior of university management. The quality of CMT decision-making is related to the safety of tens of thousands of people inside the university, which makes teachers’ and students’ expectations of university leaders grow, and also puts great pressure on the decision-making process. To pursue higher decision-making quality, CMT members have to change some of their usual leadership behaviors and pay more attention to coordinating internal communication, resolving conflicts among members, and boosting team morale.

In this environment, CMT members practice paternalistic leadership as well as contingency leadership and must adopt different leadership behaviors at the right time to promote team cohesion. This must occur to effectively improve the frequency and effect of information exchange in the decision-making process, to stimulate the initiative and participation of other team members in the CMT process, and to improve the quality of collective decision-making. According to the findings of this study, when CMT members practice paternalistic leadership, the usual simple authoritarian leadership model is no longer effective. Especially in crises, continuous high-level authoritarian leadership will alienate the relationship between leaders and other CMT members, thereby weakening their influence. However, according to the results of the surveyed university, CMT members did not completely reject the behavior of authoritarian leadership but practiced it within a limited scope to ensure that authoritarian leadership had a constraining effect on team members and subordinates, as well as to protect the effective execution of decisions that could be retained without causing serious consequences to team results.

In addition to authoritarian leadership, CMT’s practice of benevolent leadership and moral leadership still has a positive role in promoting the quality of crisis decision-making, whether directly or indirectly through conflict management. In the context of Chinese culture, the power of university leaders comes from the empowerment of the organization. In addition to the constraints imposed by the organizational system on the leader’s behavior, the self-restraint exerted by the leader’s moral sense is more dependent. This leads to the phenomenon that the organization or subordinates expect leaders to be even more ethical than they expect leaders to be competent.

This shows that the dominant role of moral leadership in paternalistic leadership can only be played by virtuous leaders who correctly play the positive role of benevolent leadership and authoritarian leadership, while leaders of low moral character can only reflect the negative role of leadership, thus leading to organizational failure. For example, when the university CMT members perceive other members, especially the key decision makers when a high level of moral leadership, as tending to internalize an identity of high moral values, ideals, and goals, the process of decision making will be more supported by high ethical standards, which is conducive to the interests of the collective project to enhance the overall quality of the decision.

Benevolent leadership, which is generally regarded as a favorable factor [11,86], is a special leadership derived from traditional Chinese family culture. The benevolent behavior reflected in it is different from the emphasis of consideration in Western literature and is more reflected in the personalized care of “different between inside and outside”. Benevolent leadership in paternalistic leadership is oriented to the care of “own people”, which itself has a strong color of inequality. This explains why many scholars have found that, in some cases, the benevolent leadership level is positively correlated with organizational performance [11,23,91], but there are also negative correlations [38]. Especially in the decision-making process, the emotional trust based on the interpersonal relationship established by benevolent leadership will eventually evolve into a blind trust, leading to a one-way information flow from top to bottom [92], resulting in low decision-making quality.

In the surveyed university, CMT members’ benevolent behavior in the face of the pandemic crisis was specifically manifested as tolerance and understanding to other team members, not embarrassing others in public, encouraging them to express their personal opinions in the decision-making process, promoting cognitive conflict, and providing diverse information to improve the quality of decision-making. On the other hand, the benevolent leadership behavior of CMT is also reflected in caring for the individual and family of members or subordinates and giving timely help. This is important in the context of a pandemic and can elicit a genuine appreciation from other team members or subordinates who, in return, are willing to sacrifice their interests to share CMT members’ values, ideals, and goals.

In addition, in traditional Chinese culture, harmony is emphasized and conflicts are avoided and negated. However, this study found that conflict was more intense and pronounced in crises. In the CMT process, the contradiction between this unavoidable conflict and the pursuit of a harmonious atmosphere within the team was the focus of this research.

In this case, CMT members must stimulate and maintain a level of cognitive conflict while suppressing affective conflict as much as possible to achieve high-quality decision-making. Because of the great pressure from outside the organization, CMT members instead have reached a short-term consensus to improve the quality of decision-making. Members have temporarily let go of personal grudges and have shown less authoritarian leadership behavior, thus inhibiting the increase of affective conflicts within the team. Additionally, the high level of moral leadership practiced among the members has a significant dampening effect on the affective conflicts that have been generated. In addition, to better survive the crisis, CMT members must create a safe and stable internal atmosphere, and it is necessary to practice high-kindness leadership behavior. This atmosphere can enhance the confidence of decision-making team members, stimulate a high level of cognitive conflict, and then improve the quality of team decision-making.

Since the outbreak of the pandemic in 2019, the surveyed university has experienced wave after wave of shocks and insisted on face-to-face teaching. Under the guidance and management of CMT, no case of infection has been reported. This suggests that the surveyed university’s CMT leadership practice has high research value, and also suggests that under the pandemic threat, it is necessary to practice different connotations of paternalistic leadership contingency, play the advantages of different dimensions, and promote the CMT for decision-making factors in the process of inhibiting factors unfavorable to the decision, in the pursuit of high-quality decisions.

### Limitations and Future Research

This study has some limitations. First, because of the Chinese government’s strict pandemic prevention and control measures, the researchers had to choose a university to collect the data, potentially limiting the generality of the results. Future studies could consider remotely recruiting participants at different universities in different regions. In addition, due to a large number of variables in the crisis situation brought by the pandemic, in addition to the variables involved in this study, other factors such as stress, communication, and cohesion may have had an impact on the CMT process, which should be considered in future studies. Finally, due to the strong timeliness of crisis management, a cross-sectional study was used to obtain data for this study, which could not reflect the causal relationship between the study variables. More longitudinal study designs may be needed in the future to discuss causal relationships between these study variables.

## Figures and Tables

**Figure 1 ijerph-19-11697-f001:**
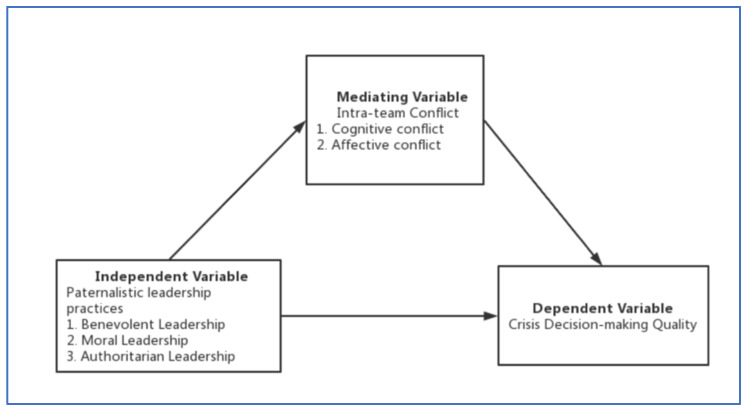
Theoretical and Conceptual Framework of This Study.

**Table 1 ijerph-19-11697-t001:** Validity and Reliability Analysis of Research Instrument.

Research Instruments		Items	Factor Loadings	Cronbach’s Alpha
Main-Scale	Sub-Scale			
Crisis Decision Quality (DQ)	Quality	1	0.93	0.93
		2	0.97	
		3	0.93	
Paternalistic Leadership (PL)	Benevolent Leadership (BL)	1	0.83	0.95
		2	0.90	
		3	0.85	
		4	0.87	
		5	0.87	
		6	0.88	
	Authoritarian Leadership (AL)	1	0.83	0.96
		2	0.87	
		3	0.83	
		4	0.87	
		5	0.91	
		6	0.87	
		7	0.85	
		8	0.88	
		9	0.85	
	Moral Leadership (ML)	1	0.71	0.93
		2	0.84	
		3	0.77	
		4	0.79	
	Total			0.80
Intra-team Conflict	Cognitive Conflict (CC)	1	0.92	0.87
		2	0.94	
		3	0.78	
	Affective Conflict (AC)	1	0.96	0.93
		2	0.91	
		3	0.91	
	Total			0.83

**Table 2 ijerph-19-11697-t002:** Means, Standard Deviations, and Correlations of Study Variables (n = 312).

		Mean	SD	1	2	3	4	5
1	AL	2.70	1.16					
2	BL	3.82	1.02	−0.32 **^,1^				
3	ML	3.81	0.79	−0.47 **	0.65 **			
4	CC	3.38	0.58	−0.33 **	0.47 **	0.53 **		
5	AC	2.46	1.03	0.23 **	−0.35 **	−0.28 **	0.14 *	
6	DQ	2.44	0.82	−0.38 **	0.60 **	0.58 **	0.61 **	−0.26 **

^1^ * *p* < 0.05, ** *p* < 0.01.

**Table 3 ijerph-19-11697-t003:** Hierarchical Regression Analysis: Control Variables, Paternalistic Leadership, Intra-team Conflict, and Decision-making Quality (n = 312).

	Model 1	Model 2	Model 3	Model 4
	**DQ**	**CC**	**AC**	**DQ**
	** *β* **	** *β* **	** *β* **	** *β* **
*Control variables*				
Level of education				
Master or above (vs. Bachelor’s degree)	0.00	0.24 **^,1^	0.51 **	0.02
Professional bg.				
Art (vs. Eco. & Mgt.)	0.06	−0.50 **	0.02	0.27 **
Law (vs. Eco. & Mgt.)	−0.35 **	−0.57 **	−0.24 **	−0.18 **
Agronomy (vs. Eco. & Mgt.)	−0.18 **	−0.15 *	0.52 **	0.00
Medical (vs. Eco. & Mgt.)	−0.06	−0.30 **	−0.02	0.05
Length of service				
<5 years (vs. >20 years)	−0.02	−0.24 **	−0.12 *	0.05
6–10 years (vs. >20 years)	0.15 **	0.05	−0.04	0.12 **
11–15 years (vs. >20 years)	−0.26 **	−0.16 **	−0.44 **	−0.30 **
15–20 years (vs. 20 years)	0.41 **	0.18 **	−0.29 **	0.27 **
Type of position				
Faculty (vs. Administration)	0.20 **	0.02	−0.01	0.19 **
Research Ins. (vs. Administration)	−0.10 **	0.18 **	0.05	−0.16 **
*Subordinate info.*				
Level of education & Gender				
Master or above (vs. Bachelor’s degree)	−0.06 *	−0.05	−0.02	−0.04 *
Female (vs. Male)	0.05 *	0.13 **	0.10 *	0.02
*Study variables*				
Authoritarian Leadership (AL)	−0.09 **	0.02	0.19 **	−0.05
Benevolent Leadership (BL)	0.19 **	0.26 **	−0.05	0.08 *
Moral Leadership (ML)	0.26 **	0.09	−0.20 **	0.18 **
Cognitive Conflict (CC)				0.41 **
Affective Conflict (AC)				−0.23 **
*R* ^2^	0.81	0.60	0.48	0.87
Adj. R^2^	0.80	0.58	0.46	0.86
*F*	79.86 **	28.05 **	17.26 **	105.21 **

^1^ * *p* < 0.05, ** *p* < 0.01.

**Table 4 ijerph-19-11697-t004:** Mediation Effect Test Results.

Mediation Paths	Results	c	a × b	c′	Formula	Percentage
		Total Effect	Mediation Effect	Direct Effect		
AL—CC—DQ	Non-significant	−0.059	0.007	−0.035	-	0%
AL—AC—DQ	**Fully mediation**	−0.059	−0.031	−0.035	-	100%
BL—CC—DQ	**Partial mediation**	0.155	0.084	0.062	a × b/c	54.32%
BL—AC—DQ	Non-significant	0.155	0.009	0.062	-	0%
ML—CC—DQ	Non-significant	0.27	0.038	0.184	-	0%
ML—AC—DQ	**Partial mediation**	0.27	0.048	0.184	a × b/c	17.75%

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
