# Peer review of "Mediating Role of Intra-Team Conflict between Paternalistic Leadership and Decision-Making Quality among China University’s CMT during COVID-19"

_ijerph, 2022, doi:10.3390/ijerph191811697_

Round 1
Reviewer 1 Report
The manuscript is very interesting.
- The introduction is exhaustive and the literature used is correct. The theses are derived from literature. The bibliography is extensive.
- The research procedure is accurately described. Reliability and credibility of research tools has been tested and described.
- The result is presented in a comprehensive and correct manner.
- The discussion and conclusions are correct.
- I propose to describe the research limitations.
Author Response
Response to Reviewer 1 Comments
Dear Review Expert,
First of all, thank you very much for your support and positive comments on our work. We attach great importance to every valuable suggestion you have made on our manuscript and have revised it accordingly. Your suggestion reminded us to add important information missing from the manuscript and helped us to further refine our research.
We have described the limitations of the study in the revised manuscript. You can find them in the conclusion section of the revised version I submitted to you in the attachment. We have marked the revised content with the "tracking" function.
We are looking forward to receiving your recognition again and extend our sincere greetings to you.

Reviewer 2 Report
The topic sounds interesting and the paper is well organized. The paper provides clarity, novelty, and contribution to the area of research and it demands a great level of in-depth understanding of the subject and a well-structured arrangement of discussions and arguments. There is one point about the discussion section need to provide information about research gaps and future perspectives: what is the paper topic added to the field, the limitation, and the further research projects.
Author Response
Response to Reviewer 2 Comments
Dear review expert,
First of all, thank you very much for your recognition and positive comments on our work. We thank you very much and attach great importance to your suggestions for improvement, and have made corresponding modifications carefully.
You can find our revisions in the latest revised manuscript that we have uploaded to you: we have added information about the research gap at the beginning of the conclusion section and "Limitations and Future Research" at the end of the conclusion section.
In addition, under your suggestion, we read the full text carefully and corrected some minor grammatical/diction errors. To make it easier for you to find our revisions, we used Word's "trace" feature to highlight them.
Finally, we sincerely thank you again for your help in further refining our research.
